# Multistaged discharge constructing heterostructure with enhanced solid-solution behavior for long-life lithium-oxygen batteries

Shu-Mao Xu [1], Xiao Liang[1], Xue-Yan Wu[1], Shen-Long Zhao[2], Jun Chen [2]*, Kai-Xue Wang [1]* & Jie-Sheng Chen[1]*

Inferior charge transport in insulating and bulk discharge products is one of the main factors resulting in poor cycling stability of lithium–oxygen batteries with high overpotential and large capacity decay. Here we report a two-step oxygen reduction approach by pre-depositing a potassium carbonate layer on the cathode surface in a potassium–oxygen battery to direct the growth of defective film-like discharge products in the successive cycling of lithium–oxygen batteries. The formation of defective film with improved charge transport and large contact area with a catalyst plays a critical role in the facile decomposition of discharge products and the sustained stability of the battery. Multistaged discharge constructing lithium peroxide-based heterostructure with band discontinuities and a relatively low lithium diffusion barrier may be responsible for the growth of defective film-like discharge products. This strategy offers a promising route for future development of cathode catalysts that can be used to extend the cycling life of lithium–oxygen batteries.

[1] Shanghai Electrochemical Energy Devices Research Center, School of Chemistry and Chemical Engineering, Shanghai Jiao Tong University, Shanghai 200240, China. [2] Department of Bioengineering, University of California, Los Angeles, CA 90095, USA. *email: jun.chen@ucla.edu; k.wang@sjtu.edu.cn; chemcj@sjtu.edu.cn

Li–O$_2$ batteries have triggered worldwide interest in energy storage system in the last decade owing to an increasing demand for high-density energy storage for electric vehicle applications[1–3]. However, the wide-range application of this battery technology is likely to be shadowed by its low round-trip efficiency, poor cycle life, and low power density[3,4]. Limited electron transfer in bulky Li$_2$O$_2$ discharge product is one of the main reasons for the difficulty to completely decompose large particles of Li$_2$O$_2$ with sufficiently fast charge transfer[4–6] and sustainably react at the electrodes with superior durability[7,8]. Directing the growth of one-dimensional rod-like[6] or film-like[4,9–12] amorphous discharge products instead of bulky toroid Li$_2$O$_2$ with enhanced charge transport is extensively adopted to accelerate and sustain the electrode reactions for high-performance Li–O$_2$ batteries. In addition to the morphology manipulation, the introduction of defects by element doping[13–16] is also utilized to enhance the charge transport and expedite the oxygen evolution reaction kinetics.

Off-stoichiometric Li$_{2-x}$O$_2$ phase ($0 < x < 1$) with Li vacancies was predicted to be metallic with enhanced electron conductivity by first-principles calculations[13,17,18]. The Li-deficient Li$_{2-x}$O$_2$ is generally believed to form during charge upon the extraction of Li$^+$ from crystalline Li$_2$O$_2$ with the release of electrons[19–21]. The formation of Li$_{2-x}$O$_2$ with faster decomposition rate than commercial Li$_2$O$_2$ powder implies enhanced charge transport and facile decomposition of electrochemically grown Li$_{2-x}$O$_2$[22,23]. Therefore, tailoring the formation of off-stoichiometric Li$_{2-x}$O$_2$ with enhanced charge transport by surface modification is of great significance to achieve Li–O$_2$ batteries with high cycling stability. Inspired by favorable Li transport through Li$_2$CO$_3$ solid electrolyte interphase (SEI) films in typical Li-ion batteries with long-term stability[24–26], surface engineering of a coating layer with feasible Li$^+$ interstitials as diffusion carriers is proposed to protect the surface of cathode catalysts and induce the formation of defective discharge products for long-life Li–O$_2$ batteries.

Herein, a two-staged discharge approach to construct Li$_2$O$_2$-based heterostructure paves a way to enhance charge transport in discharge products and improve the durability of Li–O$_2$ battery. Considering the insulator nature of most Li$_2$CO$_3$ products[27,28], a layer of K$_2$CO$_3$ instead of Li$_2$CO$_3$ is first deposited on the cathode surface in a K–O$_2$ battery. Successive discharge in Li–O$_2$ battery epitaxially deposits Li$_2$O$_2$ discharge products on the exterior surface of K$_2$CO$_3$, inducing the formation of built-in field around the junction and promoting the charge transfer and the formation of defects at the interface/surface[29–31]. Moreover, after multi-staged discharge, tailored film-like instead of large toroidal discharge products is deposited on the surface of cathode, suppressing the passivation/clogging of the cathode. The resulting PVP-C@LDO cathode can be operated in O$_2$ over 70 cycles (over 175 d) without distinct capacity decay at a limited specific capacity of 3000 mAh g$^{-1}$. Density functional calculation reveals the migration of Li$^+$ through the heterojunction to K vacant sites in K$_2$CO$_3$ (201) with energy barrier of 0.58 eV. These results benefit the design of cathode architecture by two-staged discharge to construct heterostructure for long-life Li–O$_2$ batteries.

## Results

### Synthesis of binder-free air catalysts.
As illustrated in Fig. 1a, a composite of carbonaceous coatings onto layered double oxide (LDO) nanoflakes is prepared by thermal decomposition of polyvinyl pyrrolidone (PVP) as nitrogen and carbon source to deposit on Ni foam with pre-loaded CoFe layered double hydroxides (LDH) arrays, denoted as PVP-C@LDO. Constructing building blocks of two-dimensional carbon nanosheet and LDO with embedded nanocrystals is beneficial for the integration of good chemical reactivity of LDO and superior electrical

conductivity of carbon nanosheet with enhanced catalytic activity. Scanning electron microscopic (SEM) image showed that CoFe LDH nanoflake arrays were grown vertically onto nickel foam with a flower-like morphology (Supplementary Fig. 1a). XRD pattern of as-obtained nanoflake arrays exhibits a double relationship between the basal and second-order diffractions (Fig. 1b), characteristic for the LDH structure. The layered spacing of CoFe LDH arrays calculated based on the (003) peak at 11.5° is ~7.7 Å, consistent with the spacing measured by high resolution transmission electron microscopy (HRTEM) (Supplementary Fig. 1c). After chemical vapor deposition (CVD), the lamellar morphology of LDH could also be maintained in PVP-C@LDO (Fig. 1b, c). CoFe LDH arrays on Ni foam without carbon were also calcined under the same conditions to fabricate CoFe LDO for comparison. Different from the smooth surface of CoFe LDO, the surface of PVP-C@LDO nanoflakes is decorated with plenty of particles in the average size of ~100 nm and covered with carbon layers (Fig. 1d, e). XRD analysis reveals that PVP-C@LDO is composed of CoFe alloy, Co, and graphitic carbon (Fig. 1b). The existence of Co and Fe is further demonstrated by Co 2p and Fe 2p X-ray photoelectron spectroscopic (XPS) spectra of PVP-C@LDO (Supplementary Fig. 2). HRTEM observation reveals the (110) facet of CoFe alloy of these particles and (002) facet of multilayered partially graphitic carbon on the outside layer (Fig. 1f). The elemental mapping of these nanoparticles further demonstrates the formation of CoFe alloy (Fig. 1g and Supplementary Fig. 3).

### Activity and durability of cathode catalysts.
The electrochemical properties of PVP-C@LDO as bifunctional catalysts for oxygen electrodes were evaluated at a constant current density of 100 mA g$^{-1}$ with a cut-off capacity of 1.3 mAh cm$^{-2}$ (based on the total mass of catalysts). A distinct discharge plateau is observed in the potential range of 2.8–2.6 V upon cycling in the profiles of PVP-C@LDO (Fig. 2a). PVP-C@LDO could deliver a specific discharge capacity of 4333 mAh g$^{-1}$, which means that the Li–O$_2$ battery based on PVP-C@LDO cathode was operated reversibly for over 86 h per cycle. At a cut-off voltage of 2.25 V, CoFe LDO could only deliver a specific discharge capacity of 1250 mAh g$^{-1}$ (Fig. 2b). The discharge of CoFe LDO exhibits slope-like profiles with distinct kinetic overpotential behavior (Fig. 2b). The delayed voltage response existed extensively at the start of discharge profiles of many cathode catalysts in the successive cycles[32–34], which might be closely associated with the Li$^+$/e diffusion in solid or quasi-solid discharge products[35]. The linear voltage variation could be observed in the deep discharge profiles of Ir-decorated reduced graphene oxides (Ir/GO)[36], glassy carbon[37], or branched carbon nanofibers[38]. Viswanathan et al.[37] proposed a metal–insulator–metal charge transport model to explore the electron transfer through Li$_2$O$_2$ film on glassy carbon. The above linear voltage drop was ascribed to the ohmic polarization induced by the inhibition of charge transport when the surface of the catalyst was completely covered by Li$_2$O$_2$ discharge products with poor electron conductivity. Derivation of dynamic equations further revealed the relationship between linear voltage variation and the disproportionation of LiO$_2^*$ on the catalyst surface[35,39]. The initial charge profiles of PVP-C@LDO and CoFe LDO exhibit typical three regions distinguished from distinct difference in slope (Fig. 2a, b). The first charge voltage plateau at low voltage (~3.3 V vs Li/Li$^+$) is related to the oxidation of the superoxide-like phase on the surface of Li$_2$O$_2$ with the coexistence of two phase[40,41]. The following slope-like charge profile is associated with the solid-solution delithiation to form off-stoichiometric Li$_{2-x}$O$_2$[42,43]. The second charge voltage plateau at 3.7–4.2 V is associated with the oxidation of Li$_2$O$_2$ cores with the insulating property. Compared

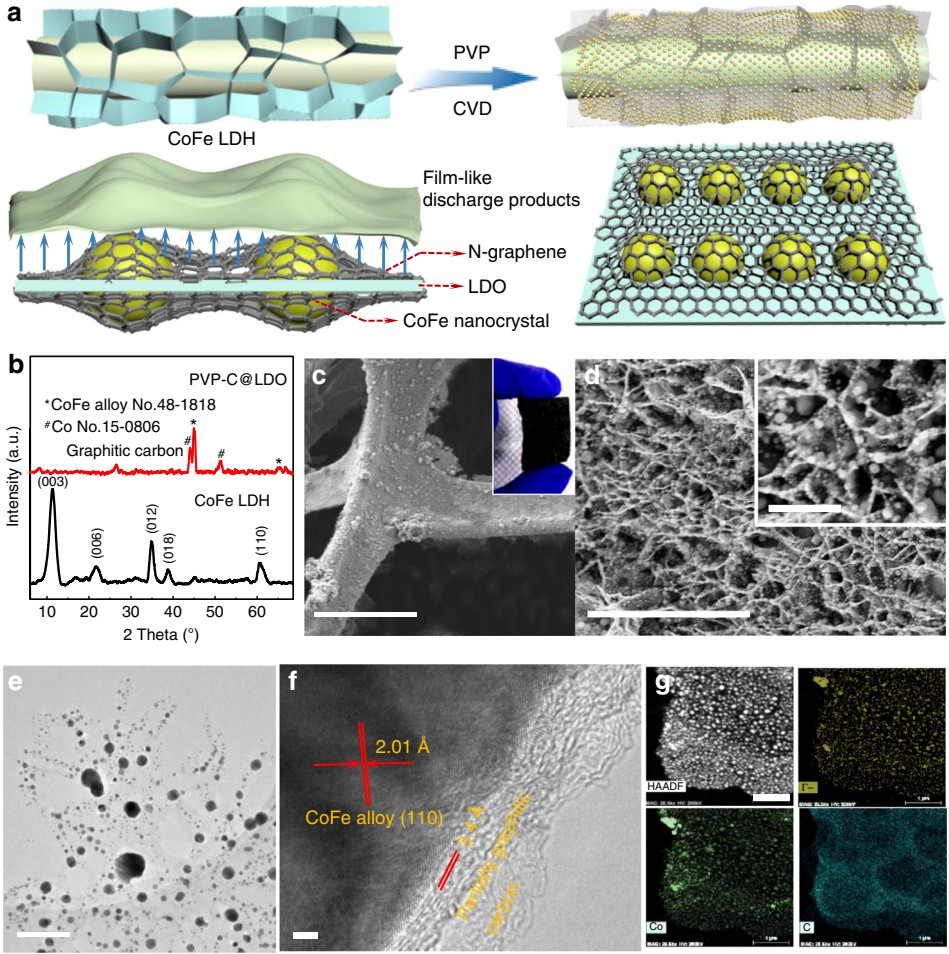

**Fig. 1 Synthesis and analysis of the cathode catalysts. a** Schemes for synthesis and the structure of PVP-C@LDO. **b** X-ray diffraction (XRD) patterns of CoFe LDH and PVP-C@LDO. **c**, **d** Scanning electron microscopic (SEM) image of PVP-C@LDO. Scale bars, 100 and 5 μm. Inset in **d** with scale bar of 1 μm. **e**, **f** Transmission electron microscopic (TEM) and high-resolution TEM (HRTEM) images of PVP-C@LDO. Scale bars, 500 and 2 nm. **g** Elemental mapping images of PVP-C@LDO. Scale bar, 1 μm.

with CoFe LDO with large capacity decay upon cycling (Fig. 2b), PVP-C@LDO exhibits enhanced cycling stability. However, the cycles of PVP-C@LDO are also limited, and fast-increasing charge overpotential is observed upon cycling. Co, Fe K-edge X-ray absorption fine structure (XAFS) spectra revealed the increase of valence states of Co and Fe in PVP-C@LDO after first charge (Supplementary Fig. 4), which might result in the change of electrochemical performance and the fast-increasing charge overpotential upon cycling. To address this issue for long-life Li–O₂ batteries, a two-step oxygen reduction approach is developed. At stage I, a discharged layer is deposited on the cathode surface in K–O₂ battery as cathode surface protective layer. The successive discharge in Li–O₂ battery at stage II constructs heterostructure with band discontinuities and interfacial transition beneficial for the formation of defective discharge products. Figure 2c, d shows the voltage profiles of PVP-C@LDO and CoFe LDO upon two-staged discharge in K–O₂ and Li–O₂ batteries. PVP-C@LDO could deliver a higher discharge capacity than CoFe LDO at stage I, while that is opposite at stage II. The weight of PVP-C@LDO electrodes after discharge at stage I is larger than that of initial electrodes. The cycles of PVP-C@LDO and CoFe LDO after two-staged discharge are fixed at a specific capacity of 1.3 mAh cm⁻² in 2.25–4.2 V. Upon cycling, both PVP-C@LDO and CoFe LDO exhibit slope-like profiles without distinct charge voltage plateau. PVP-C@LDO can be operated reversibly over 70

cycles with a specific capacity of ~3000 mAh g⁻¹ (Fig. 2e), while CoFe LDO can be operated reversibly without distinct capacity decay up to 60 cycles (Fig. 2f). The cycling stabilities of PVP-C@LDO and CoFe LDO are both greatly improved.

The deposition of discharge products on the cathode surface significantly affected the cathode/electrolyte interfacial resistance and the charge-transfer resistance on the cathode surface. Discharged PVP-C@LDO at stage II exhibits smaller cathode/electrolyte interfacial and charge transport resistance than discharged cathode in conventional Li–O₂ batteries (Fig. 2g). The cathode/electrolyte interfacial resistance of discharged PVP-C@LDO at stage II is less than half of that in conventional Li–O₂ batteries. Relative element fitting parameters in the equivalent circuit are detailed in Supplementary Table S1. Compared with pristine cathode operated in O₂, the cyclic voltammetric (CV) curve of PVP-C@LDO after discharge at stage I exhibits no distinct anodic peak (Fig. 2h), which is consistent with the disappearance of the first charge plateau in galvanostatic cycling profiles of PVP-C@LDO after stage II (Fig. 2e). The first charge voltage plateau is associated with the oxidation of LiO₂* on the surface of Li₂O₂[40,41]. The introduction of the repository layer for Li⁺ migration inducing the disappearance of the first charge plateau upon cycling indicates the feasible generation of off-stoichiometric Li₂₋ₓO₂ instead of intermediate LiO₂* with distinct solid-solution behavior upon cycling (Fig. 2i).

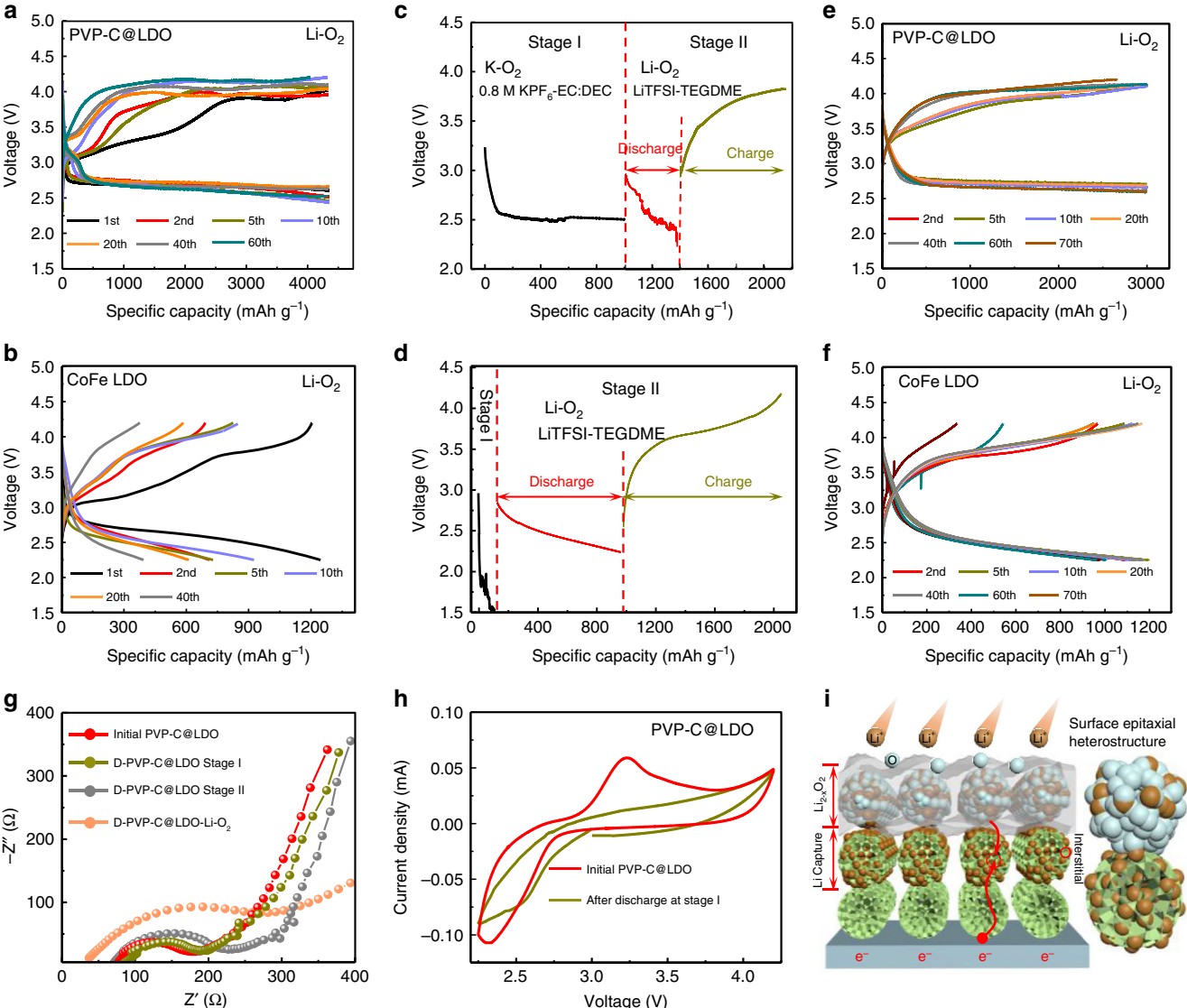

**Fig. 2 Electrochemical performance. a**, **b** PVP-C@LDO and CoFe LDO-based conventional Li–O$_2$ battery. **c**, **d** Multistaged pretreatment of PVP-C@LDO and CoFe LDO cathodes. Stage I: PVP-C@LDO/CoFe LDO|0.8 M KPF$_6$-EC: DEC (1:1)|K. Stage II: PVP-C@LDO/CoFe LDO|1 M LiTFSI-TEGDME|Li. **e**, **f** Cycling performance of PVP-C@LDO and CoFe LDO after multistaged discharge. Current densities are 100 mA g$^{-1}$. **g**, **h** Nyquist plots and cyclic voltammetric (CV) curves of PVP-C@LDO before and after discharge at a stage I and stage II. **i** Schematic representation of the formation of film-like Li$_{2-x}$O$_2$ after multistaged discharge.

**Multistaged discharge products**. To explore the effect of the deposition of multilayers on the performance of Li–O$_2$ batteries, the properties of discharge products at different stages were further characterized. C 1 s XPS spectrum of initial PVP-C@LDO exhibits a characteristic π–π* transition satellite with extended delocalized electrons at 292.7 eV (Supplementary Fig. 5). After discharge at stage I, characteristic K 2p XPS peaks with spin-orbit splitting of 2.8 eV appeared, indicating the formation of discharge products on the catalyst surface containing K. N 1 s and O 1 s XPS spectra of PVP-C@LDO after discharge at stage I show enhanced shoulder peaks of pyridinic N and Co/Fe-O (Fig. 3a, b). The relative content of CO$_3^{2-}$ component increased after discharge at stage I and further increased at stage II. XRD showed the deposition of hexagonal K$_2$CO$_3$ (JCPDS No. 27-1348) on discharged PVP-C@LDO at stage I with (201) preferential orientation (Fig. 3c). Distinct diffraction peaks of K$_2$CO$_3$ can also be found in the XRD patterns of discharged Super P cathodes over 8 h at stage I (Supplementary Fig. 6). No evidence of other potassium oxides, such as K$_2$O$_2$, KO$_2$

or K$_2$O could be detected. The existence of CO$_3^{2-}$ and the absence of the KO$_2$ and K$_2$O$_2$ were further revealed by Raman spectra of discharged PVP-C@LDO and Super P electrodes at stage I (Supplementary Fig. 7). Fourier transform infrared spectroscopy (FTIR) spectra of the discharged PVP-C@LDO at stage I and stage II further demonstrated the existence of K$_2$CO$_3$ (Fig. 3d). $^{13}$C-nuclear magnetic resonance (NMR) spectrum of Super P electrodes at stage I revealed the significant presence of K$_2$CO$_3$ ($\delta = 168.4$ ppm) (Supplementary Fig. 8). A relatively small amount of CH$_3$OK ($\delta = 58.0$ ppm) and –(CH$_2$CH$_2$O)$_n$– ($\delta = 69.7$, 67.6 ppm) could also be observed. These two species can be further identified in $^1$H-NMR spectrum (CH$_3$OK, $\delta = 3.33$ ppm; –(CH$_2$CH$_2$O)$_n$–, $\delta = 3.56$, 3.62 ppm). Based on the detected side products, the reaction mechanism related to the K$_2$CO$_3$ generation could be proposed based on the reductive cleavage of C–O bonds of EC and DEC by solvated electrons (Supplementary Fig. 8c). The mass spectra of discharged Super P, PVP-C@LDO and CoFe LDO at stage II revealed the existence of KLi$_2$CO$_3^+$, KLi$_2$O$_2^+$, K$_2$Li$_2$O$_3^+$

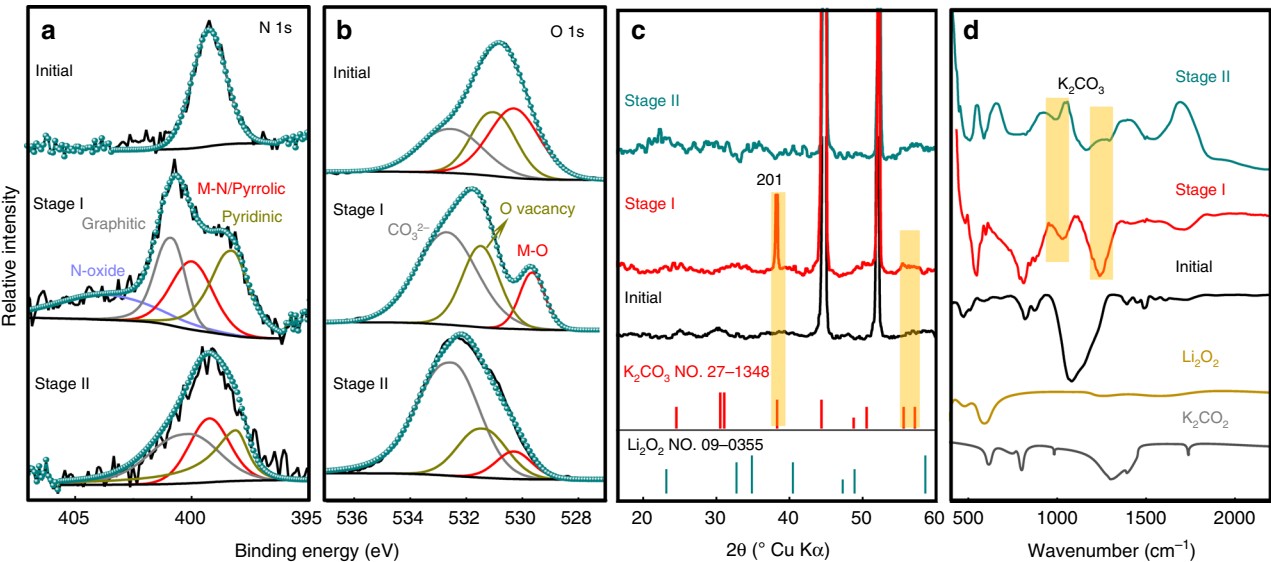

**Fig. 3 Multistaged discharge products. a, b** N 1 s and O 1 s X-ray photoelectron spectroscopy (XPS) spectra of initial PVP-C@LDO and discharged PVP-C@LDO at stage I and stage II. **c, d** XRD patterns and Fourier transform infrared spectra of initial PVP-C@LDO and discharged PVP-C@LDO at stage I and stage II.

fragments, attributable to the construction $Li_2O_2/K_2CO_3$ heterostructure (Supplementary Fig. 9).

Morphologies of initial discharge products on the surface of PVP-C@LDO and CoFe LDO cathodes at different stages were further investigated by SEM and TEM observations. Large particles with a size of ~200 nm are well dispersed on the surface of PVP-C@LDO after discharge at Stage I (Fig. 4a–c). Distinct lattice fringes with spacings of approximately 2.3 Å were detected for these cubic particles, attributable to the (201) facet of $K_2CO_3$ (Fig. 4c). SEM image of PVP-C@LDO after first discharge in traditional Li–O$_2$ battery showed typical toroid-shaped discharge products dispersed on the cathode surface (Fig. 4d). Further observation by HRTEM showed the distinct lattice fringes ascribable to $Li_2O_2$ (101) facet (Supplementary Fig. 10). However, the surface of discharged PVP-C@LDO in Li–O$_2$ battery at stage II was almost completely covered with a film-like coating (Fig. 4e). The existence of film-like discharge products could also be found in discharged CoFe LDO and Super P electrodes at stage II (Supplementary Figs. 11 and 12). Elemental mapping revealed the dispersion of K, C, and O in the outside layer of discharge products (Fig. 4f). Further observation of discharged PVP-C@LDO at stage II by HRTEM revealed the presence of heterojunction in discharge products (Fig. 4g). Distinct lattice spacings of 2.5 and 2.3 Å could correspond to (101) facet of hexagonal $Li_2O_2$ and (201) facet of $K_2CO_3$, respectively. After charge at stage II, $Li_2O_2$ grain crystal on the primary $K_2CO_3$ particles disappeared (Fig. 4h).

**Heterostructure with enhanced charge transport.** To explore the effect of heterostructure on the performance of Li–O$_2$ batteries, commercial $K_2CO_3$ was directly added into electrolyte or mixed with Super P to prepare the oxygen electrodes for comparison. When adding $K_2CO_3$ in electrolyte or coating $K_2CO_3$ with Super P on Ni foam, the capacity of Super P based Li–O$_2$ batteries was significantly reduced (Supplementary Fig. 13). Although Super P cathode after multistaged discharge delivered smaller reversible charge capacities than conventional Super P cathodes in Li–O$_2$ batteries, the durability of Super P cathode after multistaged discharge was also enhanced. To exclude the influence of fresh electrolyte on the cycling stability, the PVP-C@LDO-based Li–O$_2$ battery was disassembled after first discharge, and a new battery was rebuilt with a new Li anode,

separator, and fresh electrolyte. The first charge plateau of PVP-C@LDO-based Li–O$_2$ batteries gradually faded away, and the cycles were very limited (Supplementary Fig. 14). These results demonstrated the critical role of $Li_2O_2/K_2CO_3$ heterostructure on the durability of Li–O$_2$ batteries.

Theoretically, a prerequisite for discharge products at stage I towards ideal $Li_2O_2$-based heterostructure is to have similar crystal structure with $Li_2O_2$ to avoid lattice mismatch. For discharge at stage I in K-CO$_2$ battery with 0.8 M KPF$_6$ in EC: DEC (1: 1) as the electrolyte, XRD pattern revealed the deposition of monoclinic $K_2CO_3$ in K-CO$_2$ battery (Supplementary Fig. 15b). The cycle of Super P electrodes after multistaged discharge (stage I: K-CO$_2$ battery and stage II: Li–O$_2$ battery) is very limited with large irreversible charge capacities loss (Supplementary Fig. 15d). It is worse than that after two-staged discharge in O$_2$. For two-staged discharge in K–O$_2$ and Li–O$_2$ batteries, the common superlattice of (101) facet of hexagonal $Li_2O_2$ and (201) facet of $K_2CO_3$ with a small mismatch in both lateral dimensions indicates a translational symmetry in favor of epitaxial growth over large distances. The heterostructure could be generated after two-staged discharge with continuous deposition of $K_2CO_3$ and $Li_2O_2$. The band gap of hexagonal $Li_2O_2$ and $K_2CO_3$ using general gradient approximation (GGA) functional with $U = 6$ eV is 3.8 and 4.9 eV, respectively (Supplementary Fig. 16). The band gap could be drastically reduced after construction of the $K_2CO_3/Li_2O_2$ heterostructure. The Li migration through heterojunction was investigated by climbing-image nudged elastic band (CI-NEB) methods (Fig. 4i, j). The migration of Li through the $Li_2O_2$ (101) and $K_2CO_3$ (201) heterojunction to the K vacant sites in $K_2CO_3$ (201) is necessary to span across a transition state with an energy barrier of 0.58 eV.

Ex situ XRD analyses of Super P on carbon fiber paper (CFP) at different discharge states were utilized to evaluate the stoichiometry variation in discharge products after multistaged discharge at stage II by measuring the lattice constant change (Fig. 5a). Upon discharge at stage II, the intensity of the characteristic $Li_2O_2$ (100) peak increases gradually. The observed broad width and slight shift of the $Li_2O_2$ (100) peak before 12 h might indicate the less ordering of deposited $Li_2O_2$ discharge products. Compared with the discharged Super P electrode at 10 h in traditional Li–O$_2$ battery, an obvious shift of $Li_2O_2$ (100) peak to lower angle can be observed for discharged Super

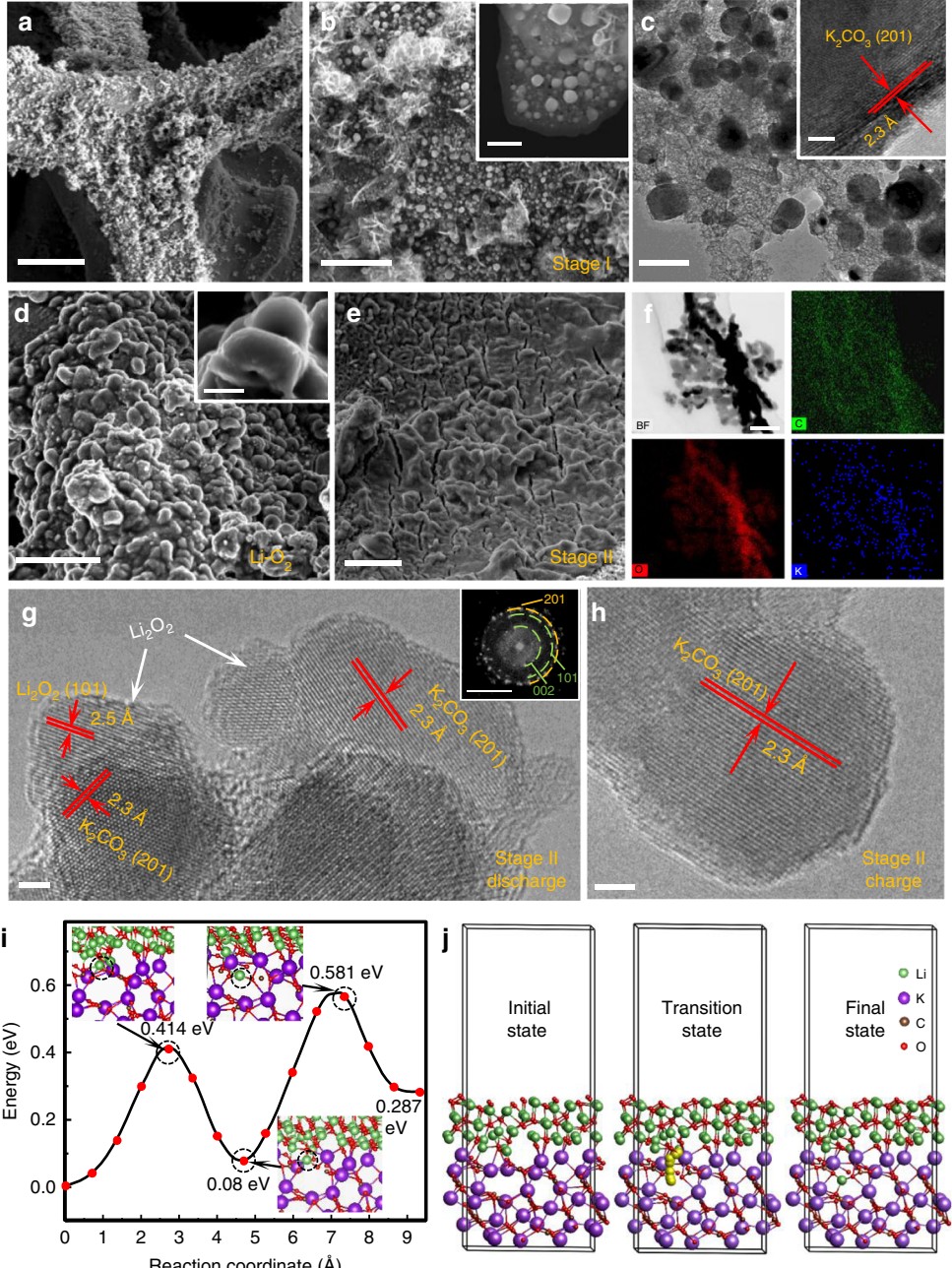

**Fig. 4 Heterostructure. a–c** SEM and TEM images of discharged PVP-C@LDO at stage I. Scale bars of **a–c** are 100 μm, 2 μm and 100 nm, respectively. Scale bars of insets in **b**, **c** are 500 nm and 2 nm, respectively. **d** SEM image of discharged PVP-C@LDO in conventional Li–$O_2$ battery. Scale bar, 5 μm. Scale bar of inset in **d** is 500 nm. **e–g** SEM, elemental mapping and HRTEM images of discharged PVP-C@LDO at stage II. **h** HRTEM image of charged PVP-C@LDO at stage II. Scale bars of **e**, **f**, **g**, and **h** are 5 μm, 50 nm, 2 nm, and 2 nm, respectively. Inset in **g** is the corresponding fast Fourier transformed image with scale bar of 5 nm$^{-1}$ (green: $Li_2O_2$; yellow: $K_2CO_3$). **i** Energy barrier profile of Li$^+$ migration through heterojunction of $Li_2O_2$ (101)/$K_2CO_3$ (201). The migration of Li$^+$ is marked as black dotted circle. **j** Li$^+$ migration pathways into K vacant site of $K_2CO_3$ (201).

P electrode at 10 h at stage II, indicating the formation of off-stoichiometric $Li_{2−x}O_2$ with larger lattice constants. It is consistent with the XRD analyses of off-stoichiometric perovskite films[44,45] and the intercalated compounds upon deinsertion of metal ions[46–48] with lattice expansion. To avoid the effect of unpaired spin from metal ions, Super P on CFP is utilized for electron paramagnetic resonance (EPR) measurements to study the defect in $Li_2O_2$ discharge products after multistaged discharge. The EPR spectra of discharged Super P at different time at stage II consist of a single quasi-symmetric resonance line

(Fig. 5b). The commercial $Li_2O_2$ powder is EPR silent. The *g* factor of discharged Super P in conventional Li–$O_2$ battery is 2.006 corresponding to the characteristic signal of $LiO_2^*$[49–51]. However, the *g* factor of discharged Super P at stage II is around 2.000. The variation of *g* factor reflects the difference of orbital contributions, spin multiplicity or spin-spin coupling[50] between EPR-active $LiO_2^*$ in conventional Li–$O_2$ battery and the two-staged discharge products after pretreatment in K–$O_2$ battery. The EPR signal intensities of discharge Super P at stage II decrease with the discharge time (Fig. 5c). The strong intensity at

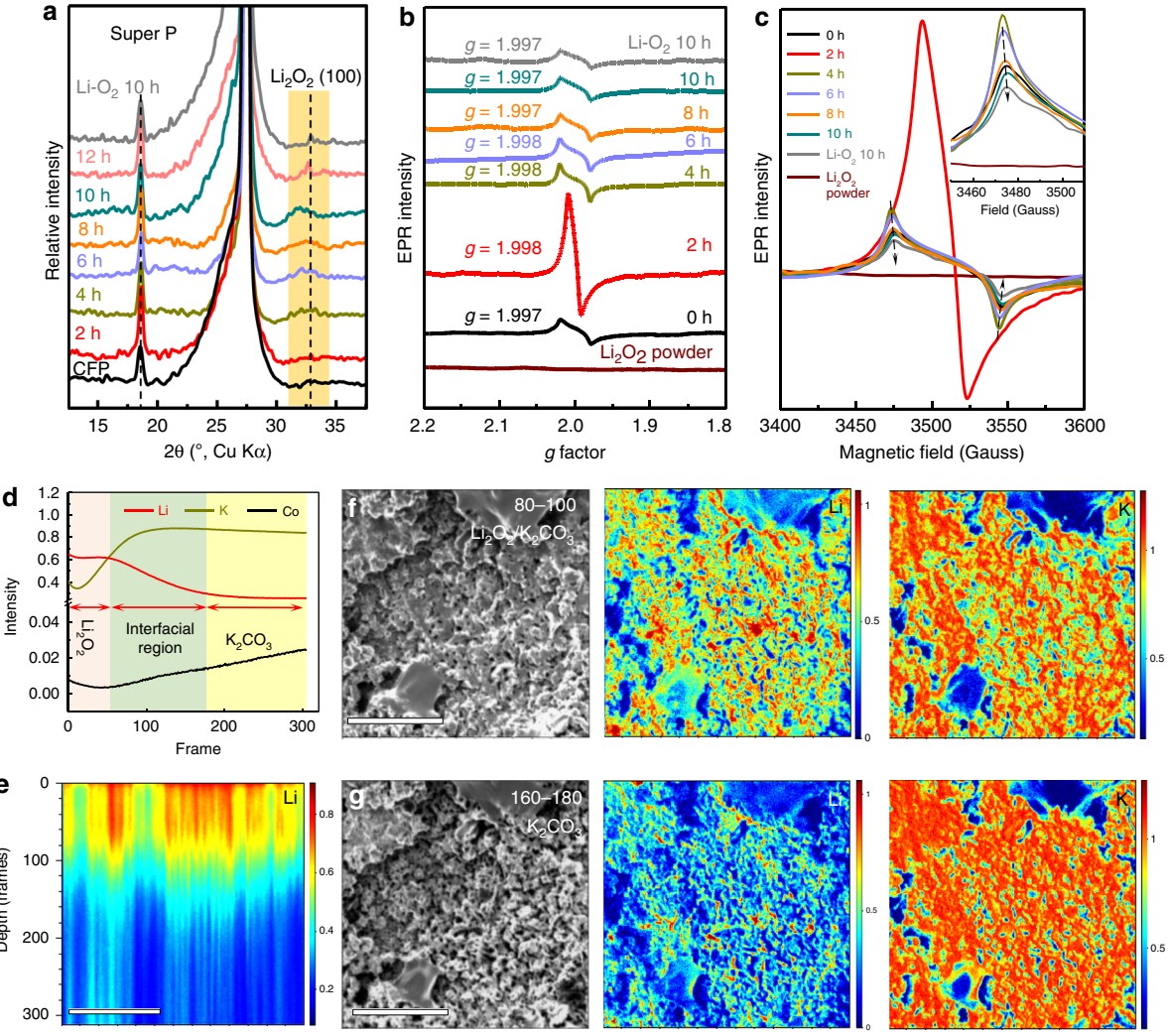

**Fig. 5 Defects in Li₂O₂. a–c** XRD patterns and electron paramagnetic resonance (EPR) spectra of conventional discharge and two-staged discharge of Super P. **d** Time-of-flight secondary ion mass spectrometric (TOF-SIMS) depth profile of discharged PVP-C@LDO at stage II. **e** The z-distribution map of Li⁺ on discharged PVP-C@LDO cathode at stage II. **f, g** TOF-SIMS mapping in the depth of 80–100 frames (~88–110 nm) and 160–180 frames (~176–200 nm). Scale bars of **e–g**, 10 μm.

2 h at stage II might be associated with the built-in field-induced charge transfer. The intensity of discharged Super P at 10 h at stage II is stronger than that of discharged Super P at 10 h in conventional Li–O₂ battery, indicative of more unpaired electrons in discharge products after multistaged discharge.

The chemical composition and distribution of K and Li throughout the discharged PVP-C@LDO at stage II is further investigated using time-of-flight secondary ion mass spectrometry (TOF-SIMS) in positive mode. The depth profile established by etching a selected area on the electrode surface under continuous Ga⁺ sputtering can be divided into several regions (Fig. 5d). In the first ~55 frames (~60 nm) of etching, the intensity plateau of Li⁺ marks the Li₂O₂ region with a relatively homogenous in-depth composition. After ~55 frames, the intensity of Li⁺ starts to decrease while that of K⁺ increases gradually to the maximum. This marks the beginning of the Li₂O₂/K₂CO₃ interfacial region. Such variation is observed until ~150 frames, at which the Li and K signals are stable, thus marking the end of this interfacial region. The z-distribution mapping image of Li⁺ on discharged PVP-C@LDO cathode at stage II reveals that species containing Li has penetrated into the surface with depth of ~150 nm (Fig. 5e). The focused ion-beam SEM (FIB-SEM) mapping images

reveal that film-like discharge products on the surface of discharged PVP-C@LDO cathode at stage II contain K and Li with clear contrast variations (Fig. 5f, g). The superposition of Li and K in the interfacial region might suggest the favorable deposition of Li₂O₂ on the substrate K₂CO₃ layer.

## Discussion

Constructing Li₂O₂-based heterostructure with band discontinuities and interfacial transition offers a promising strategy to enhance charge transport in discharge products for long-life Li–O₂ batteries. A two-step oxygen reduction approach is developed to construct K₂CO₃/Li₂O₂ heterostructure by pre-depositing a layer of K₂CO₃ on cathode surface in K–O₂ battery and epitaxial growth of Li₂O₂ after successive discharge in Li–O₂ battery. The built-in field of the heterostructure could contribute to the formation of defects at surface/interface for enhanced charge transport in discharge products.

Typically, the cycles of Super P cathodes operated in O₂ were very limited, less than 20 cycles at a fixed capacity of 0.6 mAh cm⁻². The poor durability with sluggish and unsustainable electrochemical reactions on Super P cathode is mainly originated from the insufficient charge transfer and the limited contact area between toroidal

$Li_2O_2$ and catalysts[4,8,52–54]. After multistaged discharge, Super P cathode operated in $O_2$ exhibited significantly enhanced cycling stability in spite of severe irreversible charge capacity loss (Supplementary Fig. 13). This severe irreversible charge capacity loss induced by large overpotential could be alleviated by exploiting cathode catalysts, such as PVP-C@LDO in this work with enhanced catalytic effect towards the decomposition of discharge products.

After constructing heterostructure by the two-staged discharge, both PVP-C@LDO and CoFe LDO-based Li–$O_2$ batteries exhibited distinct solid-solution behavior without distinct charge voltage plateau upon cycling. The cycling stability of PVP-C@LDO and CoFe LDO were both greatly improved (Fig. 2e, f). PVP-C@LDO-based Li–$O_2$ batteries could be operated reversibly over 70 cycles without capacity decay at a limited specific capacity of 3000 mAh $g^{-1}$. The main discharge product of $K_2CO_3$ at stage I in EC: DEC electrolyte was characterized by XPS, $^{13}$C-NMR, Raman, FTIR and mass spectra. The presence of heterostructure after two-staged discharge is revealed by HRTEM and TOF-SIMS characterization. SEM observation revealed the formation of film-like discharge products on the surface of discharged PVP-C@LDO at stage II (Fig. 4e). Compared with toroidal $Li_2O_2$, the film-like discharge products on the cathode surface with relatively large contact area can enable facile electron transport in favor of electrochemical decomposition of $Li_2O_2$[10,55]. The enhanced charge transport in $Li_2O_2$ after two-staged discharge by the construction of heterostructure is further revealed by electrochemical impedance, ex situ XRD and EPR spectroscopic analyses. $K_2CO_3$ additive had been reported to increase the cyclic performance of graphite by the formation of SEI films in Li-ion batteries[56,57]. To reveal the effect of the heterostructure on the properties of Li–$O_2$ batteries, commercial $K_2CO_3$ was directly added into the electrolyte or mixed with Super P to prepare the oxygen electrodes for comparison. When adding $K_2CO_3$ in electrolyte or coating $K_2CO_3$ on Ni foam, the capacities of Super P based Li–$O_2$ batteries were significantly reduced (Supplementary Fig. 13). These results revealed the critical role of heterostructure on the durability of Li–$O_2$ batteries.

In summary, constructing heterostructure through a two-staged discharge has been shown to improve the cycling stability of Li–$O_2$ batteries. The improvement of cycling stability might be closely associated with the enhanced charge transport in discharge products and the formation of film-like instead of large toroidal discharge products to alleviate passivation/clogging of the cathode. This new strategy opens promising avenues for the development of long-life Li–$O_2$ batteries through constructing heterostructure after multistaged discharge.

## Methods

**Cathode preparation**. 0.74 mL of concentrated ammonia (28 wt%) was added dropwise into to 40 mL 5 mmol $Co(NO_3)_2$ and 2.5 mmol $Fe(NO_3)_2$ aqueous solution to form a homogeneous solution under vigorous stirring. At a constant potential of $-2.0$ V (vs SCE), CoFe LDH was electrodeposited onto the nickel foam for 600 s in the above electrolyte with platinum foil as the counter electrode. The generated CoFe LDH on the nickel foam was rinsed twice with deionized water and then anhydrous ethanol, dried at 60 °C for 6 h. PVP-C@LDO is prepared by CVD. Typically, a quartz boat with 0.8 g PVP was placed in the upstream zone of the quartz tube. Five pieces of CoFe LDH arrays on Ni foam disks with a diameter of 12 mm were placed in the downstream zone of the tube, which is 10 cm away from the quart tube in the upstream zone. Ar was introduced to the system with a flowing rate of 150 mL $min^{-1}$. The furnace was heated to 550 °C with heating ramps of 5 °C $min^{-1}$. After 10 min, the furnace was heated to 900 °C and held for 30 min before being allowed to cool down naturally. For comparison, CoFe LDO is prepared by calcinating CoFe LDO arrays without adding PVP as carbon and nitrogen sources under the same condition.

**Electrochemistry**. The electrochemical performance was analyzed using a CR2025-type Swagelok coin battery. All of the batteries were assembled in a glove box filled with ultra-highly pure Ar using Millipore glass fiber film as the separator. Discharge at stage I was in a K–$O_2$ battery with K metal foil as anodes and 0.8 M $KPF_6$ in EC:

DEC (1: 1) as electrolyte. The discharged cathode at stage I was disassembled, washed with DEC and dried in $CO_2$ overnight. A new battery was rebuilt with the discharged cathode, Li anode and 1 M LiTFSI in the TEGDME electrolyte. For comparison, commercial $K_2CO_3$ powder mixing with Super P and PVDF (8: 1: 1) was coated on Ni foam to fabricate the cathodes for traditional Li–$O_2$ batteries. Super P cathodes were operated in $O_2$ using 1 M LiTFSI + 10 mM $K_2CO_3$ additive, and 1 M LiTFSI in TEGDME. The K–$CO_2$ batteries were assembled with Super P electrodes as cathodes, K metal foil as anodes and 0.8 M $KPF_6$ in EC: DEC (1: 1) as electrolyte. The cathodes are dried at 80 °C under vacuum for 10 h.

**Characterizations and measurements**. XRD patterns were recorded on a D/max 2550VL/PC X-ray diffractometer (Rigaku, Japan) equipped with Cu $K_\alpha$ radiation ($\lambda = 1.5418$ Å, 40 kV, 30 mA). FTIR was measured on a Spectrum 100 (PerkinElmer). XPS was performed on an AXIS Ultra DLD spectrometer (Kratos, Japan) with Al $K_\alpha$ radiation ($hv = 1486.6$ eV). Raman microprobe spectroscopy was performed on Thermo Fisher DXR, Waltham, MA, USA with an $Ar^+$ laser. The morphology of the samples was observed using a FESEM (FEI NOVA Nano SEM 230, USA). TEM and HRTEM observations were carried out on a JEM-2100F microscope operated at an acceleration voltage of 200 kV. TOF-SIMS and TOF-SEM were carried out on a TESCAN Gaia3 FESEM. EPR measurements were carried out on a Bruker ELEXSYS E580 spectrometer (9–10 GHz). The products in cathodes at stage I were extracted by $D_2O$ and tested by nuclear magnetic resonance (NMR) (Bruker, 400 MHz). The X-ray absorption (XAFS) data at the Co and Fe K-edge of the samples were recorded in transmission mode at room temperature using ion chambers (referenced samples) and fluorescence excitation mode using a Lytle detector (controlled samples) at beamline BL14W1 of the Shanghai Synchrotron Radiation Facility. During the measurement, the synchrotron was operated at 3.5 GeV and the current was controlled between 150 and 210 mA. Data processing was performed using the program ATHENA.

**DFT calculations**. First-principles electronic structure calculations were carried out using the Vienna ab initio simulation package (VASP) code[58,59] within the generalized gradient approximation (GGA) approach[60]. The generalized gradient approximation (GGA) with the Perdew–Burke–Ernzerhof (PBE) functional was used to describe the electronic exchange and correlation effects. Uniform G-centered k-points meshes with a resolution of $2\pi \times 0.04$ Å$^{-1}$ and Methfessel-Paxton electronic smearing were adopted for the integration in the Brillouin zone for geometric optimization. The simulation was run with a cut-off energy of 500 eV throughout the computations. These settings ensure convergence of the total energies to within 1 meV per atom. Structure relaxation proceeded until all forces on atoms were less than 1 meV Å$^{-1}$ and the total stress tensor was within 0.01 GPa of the target value. The energy barriers of diffusion pathways were calculated by climbing-image nudged elastic band (CI-NEB) method[61]. Hubbard corrections to the DFT Hamiltonian was introduced on the 2p orbitals of carbon and oxygen atoms to describe properly the localization of polarons using general gradient approximation (GGA) functional with U = 6 eV[28].

## Data availability

The data that support the findings within this paper are available from the corresponding author on request.

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

## Acknowledgements

This work was financially supported by the National Natural Science Foundation of China (21871177, 21673140, 21720102002) and the Opening Project of State Key Laboratory of High Performance Ceramics and Superfine Microstructure (SKL201703SIC). J.C. and S.L.Z. acknowledged University of California, Los Angeles for the startup support. The authors thank Shanghai Synchrotron Radiation Facility for the provision of beam time (BL14W1).

## Author contributions

S.M.X. and K.X.W. designed the experiment, did catalyst synthesis and characterization, battery tests and DFT calculation. X.L. participated in the batteries tests. X.Y.W. helped XRD characterization and analyses. J.S.C. and K.X.W. supervised the project. S.L.Z. and J.C. contributed to the revision of the manuscript. All authors discussed the results and commented on the manuscript.

## Competing interests

The authors declare no competing interests.

**Additional information**

