## [Peer Review File · Nature Communications]

Reviewers' comments:

Reviewer #1 (Remarks to the Author):

This manuscript develops a Li-O₂ battery with multi-staged discharge processes to form a K₂CO₃/Li₂O₂ hetero-structure on the cathode. It may be a new method for improving charge transport. However, there are many critical points unclear which should be addressed first.

(1) How to design the real cell with different electrolytes with this multi-staged discharge processes for practice application?

(2) The hetero-structure K₂CO₃/Li₂O₂ for improving charge transport should be supported by solid experimental results.

(3) The loaded K₂CO₃ could decrease the energy density of this type Li-O₂ battery. This point should be commented.

(4) The potentials of charge curves shown in Figure 2a increase with cycle. Why? Moreover, the potentials of charge curves after 10th cycle are higher than 4.0V vs Li/Li⁺ which could result in decomposition of TEGDME.

(5) Are there any K₂O₂ composited with K₂CO₃ after stage I? Are there any Li₂CO₃ composited with Li₂O₂ in stage II?

(6) The discharge curve in stage II from PVP-C@LDO shown in Figure 2c is poorer than that in Figure 2d from CoFe LDO. Why?

(7) The long cycle performance is necessary.

Reviewer #2 (Remarks to the Author):

The authors report on a very interesting strategy to avoid the conductivity limitations of Li-O₂ batteries. They pre-deposit a layer of K₂CO₃ and then use this to grow Li₂O₂ film on top which has better electronic conduction properties. This is evaluated using a range of experimental and theoretical techniques, which indicates this two-stage discharge process could be an interesting approach. In a conventional Li-O₂ cell, Li₂CO₃ layer is formed first and then Li₂O₂ grown on top and authors argue that K₂CO₃ layer owing to its lower band-gap offers a strategy to improve the overall performance. The manuscript is well-done, but I urge the authors to pay close attention to a few comments below to improve the theoretical analysis:

1) The DFT study has not used any correction for the correlation in Li₂O₂. Most prior studies have had apply some kind of hubbard correction at the GGA level (PBE) - ref. 13, etc. The authors must demonstrate that the calculated barriers do not change with respect to U.

2) What is the electronic structure of the heterostructure K₂CO₃/Li₂O₂? Is there still a large gap in Li₂O₂? Again, this has to be done with U correction. This discussion is extremely important -- see prior work in ref. 13, 16 for guidance on this analysis.

3) How does the defect phase-diagram change for the heterostructure vs standard one formed in a normal lithium-air cell. The Li₂CO₃/Li₂O₂ was originally analyzed in The Journal of Physical Chemistry Letters 3 (8), 997-1001, J. Phys. Chem. C 117, 5568-5577, etc. Can the authors compare and contrast with the K₂CO₃/Li₂O₂ heterostructure vs Li₂CO₃/Li₂O₂?

4) There should be a band diagram, at least included in the SI, for the band levels for K₂CO₃, Li₂O₂, Li₂CO₃ based on the DFT calculations. Again, please incorporate U correction to ensure this is done correctly.

V. Viswanathan

Reviewers comments:

Reviewer #1 (Remarks to the Author):

This manuscript develops a Li-O₂ battery with multi-staged discharge processes to form a K₂CO₃/Li₂O₂ hetero-structure on the cathode. It may be a new method for improving charge transport. However, there are many critical points unclear which should be addressed first.

We are highly grateful for the referee's appreciation on the novelty of enhancing charge transport in discharge products by construction of K₂CO₃/Li₂O₂ heterostructure. According to the suggestions and recommendations, careful revision has been made to the manuscript. All the corrections and amendments are highlighted in the revised manuscript.

(1) How to design the real cell with different electrolytes with this multi-staged discharge processes for practice application?

Response:

In this work, the multistaged discharge is processing in the coin cells to explore the different stage discharge performance and the different stage discharge products. For practical application, the coin cells at stage I to generate K₂CO₃ can be replaced by the three/two electrode systems just as electrodeposition/electroplating using K foil as the counter electrode, Ag/AgCl or K foil as the reference electrode, and 0.8 M KPF₆ in EC: DEC (1:1) with saturated O₂ as the electrolyte. Then, the pretreated electrodes can be washed with DME/DEC and dried under CO₂ flow for the successive Li-O₂ battery assembly.

Indeed, it will be convenient to integrate multi-staged discharge into a single system. We will further explore the integrated battery with Li-K alloy anode, hybrid electrolytes or Li salt electrolyte with K salt additives as the electrolyte in our successive research.

(2) The hetero-structure K₂CO₃/Li₂O₂ for improving charge transport should be supported by solid experimental results.

Response:

We sincerely thank the reviewer for raising this professional comment. The heterostructure of K₂CO₃/Li₂O₂ in this work is characterized by HRTEM (Fig. 4g) and TOF-SIMS in-depth sputtering mapping images (Fig. 5d-g). The Mott-Schottky plots

revealed the formation of n-n heterostructure after multistaged discharge (Supplementary Fig. S22). The enhanced charge transport is revealed by EIS (the smaller cathode/electrolyte interfacial resistance of discharged PVP-C@LDO at stage II than that in conventional Li-O₂ batteries); *Ex-situ* XRD (the presence of off-stoichiometric discharge products) and EPR (more unpaired electrons in discharge products after multistaged discharge).

To reveal the effect of the heterostructure on the properties of Li-O₂ batteries, commercial K₂CO₃ was directly added into the electrolyte or mixed with Super P to prepare the oxygen electrodes for comparison. When adding K₂CO₃ in electrolyte or coating K₂CO₃ on Ni foam, the capacities of Super P based Li-O₂ batteries were significantly reduced (Supplementary Fig. S13). These results revealed the critical role of heterostructure on the durability of Li-O₂ batteries.

The enhanced charge transport in Li₂O₂ might be induced by K₂CO₃/Li₂O₂ heterostructure with band discontinuities and relatively low Li diffusion barrier by NEB calculation. Theoretically, the formation of K₂CO₃/Li₂O₂ heterostructure with built-in field would promote the charge transfer from K₂CO₃ to Li₂O₂ and the formation of defects at the interface/surface (*Nat. Rev. Mater.* **3**, 17089 (2018); *Nat. Commun.* **5**, 5118 (2014)). Therefore, the formation of heterostructure K₂CO₃/Li₂O₂ could enhance charge transport in Li₂O₂ discharge products.

To emphasize the effect of heterostructure in charge transport of Li₂O₂ discharge products, a new section entitled “**Heterostructure with enhanced charge transport**” is added to the revised manuscript. Moreover, relative summary is added in **Discussion** as listed below.

“The enhanced charge transport in Li₂O₂ after two-staged discharge by the construction of heterostructure is further revealed by electrochemical impedance, *Ex-situ* XRD and EPR spectroscopic analyses.”

(3) *The loaded K₂CO₃ could decrease the energy density of this type Li-O₂ battery. This point should be commented.*

Response:

We sincerely thank the reviewer for carefully reviewing our work. The cycling of PVP-C@LDO cathodes in conventional Li-O₂ batteries and at stage II are both fixed at 1.3 mAh cm⁻² for comparison. At stage I, the discharge of PVP-C@LDO is fixed at a specific capacity of 0.3 mAh cm⁻². The weight of PVP-C@LDO electrodes after discharge at stage I is ~0.44 mg cm⁻², larger than that of initial electrode (~0.3 mg cm⁻²). The specific capacity of CoFe LDO at stage I is very tiny, and the weight increase after discharge at stage I of CoFe LDO is very little. Therefore, the specific

gravimetric energy density of cycled PVP-C@LDO after multistaged discharge is smaller than that of conventional PVP-C@LDO-based Li-O₂ battery, while the specific gravimetric energy density of cycled CoFe LDO after multistaged discharge is similar with that of conventional Li-O₂ battery. The following description of the specific capacity is added to the revised manuscript.

“The weight of PVP-C@LDO electrodes after discharge at stage I is larger than that of initial electrodes. The cycles of PVP-C@LDO and CoFe LDO after two-staged discharge are fixed at a specific capacity of 1.3 mAh cm⁻² in 2.25-4.2 V.”

Theoretically, according to the Faraday law, if the generated K₂CO₃ is all deposited on cathode surface at stage I, the weight of generated K₂CO₃ is ~0.77 mg cm⁻² (138.2/2×0.3×10⁻³/26.8×1000). However, the weight of PVP-C@LDO electrodes after discharge at stage I is only ~0.44 mg cm⁻². For PVP-C@LDO and Super P electrodes in conventional Li-O₂ batteries, we also found that the increased weight of electrodes after discharge is much less than the theoretical weight of deposited Li₂O₂ electrodes. It might be associated with the generated solvated intermediate/discharge products, such as solvated superoxide or peroxide; or other side reactions such as Joule heating, electrode activation.

(4) The potentials of charge curves shown in Fig. 2a increase with cycle. Why? Moreover, the potentials of charge curves after 10th cycle are higher than 4.0 V vs Li/Li⁺ which could result in decomposition of TEGDME.

Response:

We highly appreciate the reviewer’s professional comment. After first charge in conventional Li-O₂ battery, XAFS spectroscopy analyses reveal the increase of valence states of Co and Fe in PVP-C@LDO, and the presence of Co/Fe-O after 1st cycle (Fig. R1). Co, Fe could be oxidized in strong oxidation environment in O₂ upon charge, which might result in the change of electrochemical performance of cathode catalysts.

Fig. R1. Co/Fe K-edge XAFS and EXAFS of CoFe LDO and PVP-C@LDO after first charge in conventional Li-O₂ batteries. (Fig. R1 is added as Fig. S4 in **Supporting information.**)

As shown in Fig. R2, the potentials of charge curves of PVP-C@LDO after 5th cycle are found to look more like those of Co₃O₄ and CoO cathodes as reported in the literature (*Nano Lett.*, **16**, 5902-5908 (2016); *Sci. Rep.*, **5**, 8335 (2015)). The change of potentials upon cycling can also be found in MoNi alloy catalysts in Li-O₂ batteries in our group's work (Fig. R3), and the existence of MoO_x phase is also found after cycling in Li-O₂ battery, resulting from the oxidation of Mo. Therefore, the potentials change of charge curve might be associated with the oxidation of catalysts in O₂.

[Redacted]

Fig. R2. The electrochemical performance of a, PVP-C@LDO in conventional Li-O₂ battery; b, Co₃O₄ and CoN (*Nano Lett.*, **16**, 5902-5908 (2016).); c-d, CoO (*Sci. Rep.*, **5**, 8335 (2015).)

Fig. R3. The electrochemical performance of MoNi alloys in Li-O₂ batteries.

TEGDME is one of the most stable electrolytes in Li-O₂ batteries, which is stable up to 4.3 V (*Energy Environ. Sci.*, **9**, 1783–1793 (2016).) To avoid the decomposition of electrolyte, the cycles of PVP-C@LDO, CoFe LDO and Super P cathodes are fixed in 2.25-4.2 V. As shown in Fig. R4h, no distinct anodic peaks are observed in the CV curves of PVP-C@LDO based Li-O₂ batteries with TEGDME as electrolyte in 4-4.2 V, excluding the decomposition of TEGDME. The charge curves of PVP-C@LDO in Li-O₂ batteries exhibited large variation upon cycling. After two-staged discharge, although the charge overpotential cannot be effectively lowered, the cycling stability of PVP-C@LDO is greatly improved. The degradation of charge performance is suppressed. Therefore, the multistaged discharge to construct K₂CO₃/Li₂O₂ heterostructure can protect the cathode surface, which is beneficial for the improved cycling stability of Li-O₂ batteries.

Fig. R4. a,b, PVP-C@LDO and CoFe LDO based conventional Li-O₂ battery. c,d, Multistaged pretreatment of PVP-C@LDO and CoFe LDO cathodes. e,f, Cycling performance of PVP-C@LDO and CoFe LDO after multistaged discharge. g,h, Nyquist plots and CV curves of PVP-C@LDO before and after discharge at a stage I and stage II. i, Schematic representation of the formation of film-like discharge products after multistaged discharge.

(5) Are there any K₂O₂ composited with K₂CO₃ after stage I? Are there any Li₂CO₃ composited with Li₂O₂ in stage II?

Response:

We truly appreciate the reviewer's professional question. In K-O₂ battery with ether electrolyte, KO₂ was confirmed as the dominant discharge product because of its kinetic and thermodynamic stability. At stage I, in order to generate K₂CO₃ instead of KO₂, the carbonate ester (EC: DEC) with KPF₆ is selected as electrolyte in K-O₂ battery instead of the generally used DME electrolyte with KTFSI. Moreover, the discharged electrodes at stage I are washed with DEC and dried in CO₂ overnight to transform the residual KO₂ to K₂CO₃.

XRD patterns of discharged Super P cathodes over 8 h at stage I showed the existence of distinct K_2CO_3 diffraction peaks (Fig. R5). No evidence of other potassium oxides, such as K_2O_2 , KO_2 or K_2O can be seen. The mass spectra of discharged PVP-C@LDO cathodes at stage I revealed the absence of K_2O_2^+ fragment (characteristic KO_2 signal) (Fig. R5).

Fig. R5. a, XRD patterns of discharged Super P cathodes with different time at stage I. b, Mass spectrum of discharge products on 1st discharged PVP-C@LDO at stage I. (Fig. R5 is added as Fig. S6 in **Supporting information**.)

Raman spectra of PVP-C@LDO and Super P electrodes after discharge at stage I revealed the existence of CO_3^{2-} and the absence of the KO_2 and K_2O_2 (Fig. R6).

Fig. R6. Raman spectra of PVP-C@LDO and Super P electrodes after discharge at stage I. (Fig. R6 is added as Fig. S7 in **Supporting information.**)

However, in addition to the detected K_2CO_3 specie, CH_3OK ($\delta=3.33$ ppm, singlet and $\delta=58.0$ ppm), $-(CH_2CH_2O)_n-$ ($\delta=3.56, 3.62$ ppm and $\delta=69.7, 67.6$ ppm) can be identified in the 1H -NMR and ^{13}C -NMR spectra of discharge products on Super P electrodes at stage I in D_2O (Fig. R7). In Li-ion battery, solvated Li^+ in EC electrolyte gaining electron will lead to the continuous reduction of EC to generate Li_2CO_3/CO_2 (*Angew. Chem. Int. Ed.* **57**, 15002-15027 (2018).). Based on the detected side products, the reaction mechanism to generate K_2CO_3 can be proposed based on the reductive cleavage of C-O bonds by solvated electrons as shown in Fig. R7c.

Fig. R7. a, ^{13}C -NMR and b, 1H -NMR of the cathode surface layer at stage I in D_2O solution. c, Proposed mechanism of the formation of K_2CO_3 at stage I. (Fig. R7 is added as Fig. S8 in **Supporting information.**)

Theoretically, a prerequisite for discharge products at stage I towards ideal Li_2O_2 -based heterostructure is to have similar crystal structure with Li_2O_2 to avoid lattice mismatch. Neither CH_3OK (inorganic solid salt) nor $-(CH_2CH_2O)_n-$ (without conjugated structure) can construct heterostructure. For discharge at stage I in K- CO_2 battery with 0.8 M KPF_6 in EC: DEC (1: 1) as the electrolyte, XRD pattern revealed the deposition of monoclinic K_2CO_3 in K- CO_2 battery instead of the hexagonal K_2CO_3 in K- O_2 battery (Fig. R8b). The cycle of Super P electrodes after multistaged

discharge (stage I: K-CO₂ battery and stage II: Li-O₂ battery) is worse than that after two-staged discharge in K-O₂ battery and Li-O₂ battery (Fig. R8d). For two-staged discharge in K-O₂ and Li-O₂ batteries, the common superlattice of (101) facet of hexagonal Li₂O₂ and (201) facet of hexagonal K₂CO₃ with a small mismatch in both lateral dimensions is beneficial for epitaxial growth of Li₂O₂ over large distances and the construction of K₂CO₃/Li₂O₂ heterostructure.

Fig. R8. a, Cycle performance of Super P electrodes in K-CO₂ batteries. b, XRD patterns of discharged Super P at stage I in K-CO₂ battery and at stage I in K-O₂ battery. c, Two-staged discharge of Super P electrodes in stage I: K-CO₂ battery with EC:DEC as electrolyte and stage II: Li-O₂ battery with TEGDME as the electrolyte. d, Cycle performance of Super P electrodes after two-staged discharge. SEM images of discharged Super P e, at stage I (K-CO₂ battery); f, at stage II. (Fig. R8 is added as Fig. S15 in **Supporting information**.)

In conventional Li-O₂ batteries, the existence of Li₃CO₃⁺ fragment, characteristic specie of Li₂CO₃ in mass spectrum revealed the side reaction of Li₂O₂ with carbon to generate Li₂CO₃ (Fig. R9). However, the mass spectra of Super P, PVP-C@LDO and

CoFe LDO at stage II revealed no distinct Li_3CO_3^+ fragment (Fig. R9,10). The existence of $\text{KLi}_2\text{CO}_3^+$, KLi_2O_2^+ , $\text{K}_2\text{Li}_2\text{O}_3^+$ fragments is closely associated with the formation of $\text{Li}_2\text{O}_2/\text{K}_2\text{CO}_3$ heterostructure. Constructing $\text{Li}_2\text{O}_2/\text{K}_2\text{CO}_3$ heterostructure can suppress the formation of Li_2CO_3 on carbon catalyst surface in $\text{Li}-\text{O}_2$ batteries.

Fig. R9. Mass spectra of discharged products on 1st discharged Super P cathodes surface at stage II and in conventional $\text{Li}-\text{O}_2$ battery. (Fig. R9 is added as Fig. S21 in **Supporting information**.)

Fig. R10. Mass spectra of discharged products on 1st discharged PVP-C@LDO and CoFe LDO cathodes surface at stage II. (Fig. R10 is added as Fig. S20 in **Supporting information**.)

In summary, we highly appreciate the reviewer's professional question, which helped us to make extensive investigations in details to elaborate our points as well as to justify the significance of this work. And we sincerely hope that the detailed

explanations and analysis here could throw some light upon the novel aspects of this work. And we really hope that our explanation and additional experimental data in the revised manuscript could be rewarded an appreciation from the reviewer.

(6) *The discharge curve in stage II from PVP-C@LDO shown in Fig. 2c is poorer than that in Fig. 2d from CoFe LDO. Why?*

Response:

We sincerely thank the reviewer for carefully reviewing our work. PVP-C@LDO could deliver a higher discharge capacity than CoFe LDO at stage I, while that is opposite at stage II. However, the discharge of Super P at stage I and stage II showed no distinct difference. This difference could be associated with the different discharge curves with different “slopes”. The discharged PVP-C@LDO at stage I and the discharged Super P at stage I/II exhibited distinct discharge voltage “plateau”. The discharged CoFe LDO at stage I/II and first discharged PVP-C@LDO at stage II exhibited “slope-like” discharge profiles. Moreover, the “slope” of discharged PVP-C@LDO at stage II is larger than that of discharged CoFe LDO.

Theoretically, a plateau in the galvanostatic discharge profiles of Li/K-O₂ batteries demonstrates the steady-state polarization of oxygen reduction, while a “slope-like” discharge profile in Li/K-O₂ batteries is associated with the kinetic overpotential behavior originated from the second electron transfer accompanied by Li⁺/K⁺ migration in solid/quasi-solid intermediates (*Angew. Chem. Int. Ed.* **130**, 6941-6945 (2018)). For discharge at stage II, if Li⁺/K⁺ migration in the initial discharged products at stage I accompanied with electron transfer could occur, the discharge would exhibit distinct “slope-like” profile. The Li⁺/K⁺ migration in the discharge products will result in the structure transformation, such as n-doping, defects, stress release (incoherent interface/grain boundaries/particles crack), component/phase change in the substrate discharge products.

TOF-SIMS SEM image of discharged Super P at stage II after 300 frames sputtering by Ga⁺ revealed the well-dispersed particles in the substrate layer (Fig. R11), which keep in line with the initial discharge products at stage I (Fig. R12). However, TOF-SIMS SEM images of discharged PVP-C@LDO at stage II upon sputtering from 0-300 frames revealed no initial large particle-like discharge products at stage I (Fig. R11). The discharge of Super P at stage II deposited discharge products coating on the discharge products at stage I, which caused less change in the morphology of discharge products at stage I. However, the morphology variation of discharged

PVP-C@LDO at stage I is large after discharge at stage II, which might result in the “slope-like” and “shaking” curve of discharge PVP-C@LDO at stage II.

Fig. R11. TOF-SIMS SEM images of discharged a, Super P; b, PVP-C@LDO; c, CoFe LDO cathodes at stage II upon sputtering by Ga^+ in 0 frame (before sputtering), 100, 200 and 300 frames.

Fig. R12. SEM images of Super P electrodes after discharge at stage I (10 h and 14 h), at stage II, and in conventional Li-O₂ batteries. (Fig. R12 is added as Fig. S12 in **Supporting information**.)

This phenomenon can also be observed in the two-staged discharge of PVP-C@LDO in Li-O₂ batteries (Fig. R13). PVP-C@LDO electrode after 20 h discharge in Li-O₂ battery was disassembled and rebuilt with new Li anode, separator, and fresh electrolyte. The discharge of the rebuilt PVP-C@LDO-based Li-O₂ battery exhibits the “slope-like” profiles.

Fig. R13. Disassembly of the first discharged PVP-C@LDO cathode in Li-O₂ battery and reassembly of the Li-O₂ battery with new electrolyte and separator. a, Multistaged pretreatment of PVP-C@LDO in Li-O₂ batteries. b, Cycle performance of PVP-C@LDO after multistaged discharge. c, SEM images of discharged PVP-C@LDO at stage II in Li-O₂ batteries. (Fig. R13 is added as Fig. S14 in **Supporting information**.)

CoFe LDO without any carbon could only deliver a tiny discharge capacity in K-O₂ batteries at stage I. The discharge of CoFe LDO in traditional Li-O₂ batteries and at stage II exhibits both distinct “slope-like” profiles. The discharge of CoFe LDO, mixed metal oxide, in Li-O₂ batteries can be considered as the construction of Li₂O₂/metal oxide heterostructure (*J. Phys. Chem. Lett.* **4**, 3494-3499 (2013)). Therefore, the nature of the “slope-like” discharge profiles of CoFe LDO is similar with that of PVP-C@LDO at stage II.

The difference in the discharge performance of the catalysts with/without carbon might be associated with different preferential adsorption and coverage of reactants/intermediates on the activated sites accompanied by different electron transferring. This phenomenon is further revealed in our group’s recent work on sodium poly(aminobenzenesulfonate) derived carbon nanosheets with different C-S groups towards different (dis)charge performance in Li-O₂ batteries (Fig. R14).

Fig. R14. Selective (dis)charge curves of a, PABSA/LDO-300; b, PABSA/LDO-600; c, PABSA/LDO-700; and d, PAN/LDO-700 at $26.5 \mu\text{A cm}^{-2}$.

(7) *The long cycle performance is necessary.*

Response:

We truly appreciate the reviewer's suggestions. The cycle performance of PVP-C@LDO and CoFe LDO after two-staged discharge is shown in Fig. 2e,f, respectively. The resulting PVP-C@LDO cathode can be operated in O_2 over 70 cycles (over 175 d) without distinct capacity decay at a limited specific capacity of $3,000 \text{ mAh g}^{-1}$ at 100 mA g^{-1} . The cycles of PVP-C@LDO-based Li- O_2 batteries fixed at a specific capacity of 3000 mAh g^{-1} over 175 d could meet the requirement of cycling life in practical application.

The selective (dis)charge curves are shown in the manuscript to compare the cycling stability after two-staged discharge. The initial/cutoff voltage and specific capacity with cycle number are added in **Supporting Information** listed as below.

Fig. R15. Cycling performance of PVP-C@LDO and CoFe LDO a, in conventional Li-O₂ batteries; b, after two-staged discharge. (Fig. R15 is added as Fig. S22 in **Supporting information**.)

Reviewer #2 (Remarks to the Author):

The authors report on a very interesting strategy to avoid the conductivity limitations of Li-O₂ batteries. They pre-deposit a layer of K₂CO₃ and then use this to grow Li₂O₂ film on top which has better electronic conduction properties. This is evaluated using a range of experimental and theoretical techniques, which indicates this two-stage discharge process could be an interesting approach. In a conventional Li-O₂ cell, Li₂CO₃ layer is formed first and then Li₂O₂ grown on top and authors argue that K₂CO₃ layer owing to its lower band-gap offers a strategy to improve the overall performance. The manuscript is well-done, but I urge the authors to pay close attention to a few comments below to improve the theoretical analysis:

We sincerely thank the reviewer for carefully reviewing our work. And meanwhile, we are highly grateful for the reviewer's generous comments on our work as "interesting strategy", "interesting approach" and "well-done work".

1) *The DFT study has not used any correction for the correlation in Li₂O₂. Most prior studies have had apply some kind of hubbard correction at the GGA level (PBE) - ref. 13, etc. The authors must demonstrate that the calculated barriers do not change with respect to U.*

Response:

We highly appreciate the reviewer's professional suggestion. Generally, the DFT+U approach is utilized for voltage calculations to reduce self-interaction errors in *d* or *f* transition metals-based system with highly correlated electrons. Although oxygen is not attributed to a transition element, a Hubbard U on the oxygen 2p states was added in the DFT calculations (PBE+U) to localize the charge on the O₂ nearest the Li vacancy (*Energy Environ. Sci.* **7**, 720-727 (2014); *J. Phys. Chem. C* **117**, 5568–5577 (2013).) In this work, the migration barrier is calculated by CI-NEB method. For NEB calculation, previous studies have reported the convergence difficulties based on DFT+U owing to the metastability of the electronic states along the migration path (*Chem. Mater.* **13**, 6646-59 (2018).). Moreover, the converged NEB calculations within DFT+U have been reported to show insignificant difference in the barriers calculated by DFT (*Phys. Rev. B* **16**, 075112 (2011).).

As suggested by the reviewer, we have made great efforts and tried our best to calculate migration barriers in K₂CO₃/Li₂O₂ heterostructure by DFT+U with different U values and convergence precision. However, the K₂CO₃/Li₂O₂ heterostructure containing hundreds of atoms, and the disorderly arrangement of atoms at the interface resulted in large convergence difficulties based on DFT+U.

The correction with U=6 eV is introduced in the band structure calculation. Relative description is added to the revised **Experimental Section** as below.

“Hubbard corrections to the DFT Hamiltonian was introduced on the 2p orbitals of carbon and oxygen atoms to describe properly the localization of polarons using general gradient approximation (GGA) functional with U=6 eV²⁸.”

2) *What is the electronic structure of the heterostructure K₂CO₃/Li₂O₂? Is there still a large gap in Li₂O₂? Again, this has to be done with U correction. This discussion is extremely important -- see prior work in ref. 13, 16 for guidance on this analysis.*

Response:

We sincerely thank the reviewer for raising this constructive comment. The electronic structure of the $\text{K}_2\text{CO}_3/\text{Li}_2\text{O}_2$ heterostructure is shown in Fig. R16. The calculated band gap of Li_2O_2 and K_2CO_3 using GGA+U functional with $U=6$ eV is 3.8 and 4.9 eV, respectively. A drastic drop of band gap could be observed after construction of the $\text{K}_2\text{CO}_3/\text{Li}_2\text{O}_2$ heterostructure.

The following analyses are added in the revised manuscript as below.

“The band gap of Li_2O_2 and K_2CO_3 using general gradient approximation (GGA) functional with $U=6$ eV is 3.8 and 4.9 eV, respectively (Supplementary Figure S16). The band gap could be drastically reduced after construction of the $\text{K}_2\text{CO}_3/\text{Li}_2\text{O}_2$ heterostructure.”

Fig. R16. Calculated densities of states (DOS), using GGA+U with $U=6$ eV, for a, Li_2O_2 ; b, $\text{K}_2\text{CO}_3/\text{Li}_2\text{O}_2$ heterostructure; c, Li_2CO_3 ; d, K_2CO_3 . (Fig. R16 is added as Fig. S16 in Supporting information.)

3) How does the defect phase-diagram change for the heterostructure vs standard one formed in a normal lithium-air cell. The $\text{Li}_2\text{CO}_3/\text{Li}_2\text{O}_2$ was originally analyzed in *The Journal of Physical Chemistry Letters* 3 (8), 997-1001, *J. Phys. Chem. C* 117, 5568-5577, etc. Can the authors compare and contrast with the $\text{K}_2\text{CO}_3/\text{Li}_2\text{O}_2$ heterostructure vs $\text{Li}_2\text{CO}_3/\text{Li}_2\text{O}_2$?

Response:

We sincerely thank the reviewer for raising this professional question. The Li/K- O_2 batteries in this work are operated in 1 atm O_2 at room temperature. Phase diagram of

Li₂O₂/Li₂CO₃ products in typical carbon-based Li-O₂ batteries reveals the feasible existence of stable intermediates/products of Li₂O, Li₂O₂, Li₂CO₃ (Fig. R17). After multistaged discharge, phase diagram reveals the feasible newly generated intermediates/products of K₂O, K₂O₂, KO₂, K₂CO₃, KCO₃, KLiCO₃. Relative formation/decomposition energy is listed in Table. R1.

[Redacted]

Fig. R17. Phase diagram of a, K-Li-C-O and b, Li-C-O. (*APL Mater.* **1**, 011002 (2013).)

Table. R1. Stable intermediates/products in phase diagram of K-Li-C-O.

Formula	Formation/Decomposition (eV/Atom)	id
C	0	mp-568286
CO ₂	-1.78	mp-20066
K	0	mp-1184905
K ₂ CO ₃	-2.13	mp-3963
K ₂ O	-1.257	mp-971
K ₂ O ₂	-1.30	mp-2672
KC ₈	-0.03	mp-28930
KCO ₃	-1.83	mp-1106094
KLiCO ₃	-2.205	mp-562137
KO ₂	-0.98	mp-1866
Li	0	mp-135
Li ₂ CO ₃	-2.26	mp-3054
Li ₂ O	-2.07	mp-1960
Li ₂ O ₂	-1.65	mp-841
LiC ₁₂	-0.009	mp-1021323
O ₂	0	mp-12957

Li₂O₂ is found to crystallize in a hexagonal structure with lattice parameters a=b=3.19 Å, c=7.73 Å. The band gap of Li₂CO₃ (6.4 eV) is much larger than that of K₂CO₃ (4.9 eV) and Li₂O₂ (3.8 eV). Relative structure parameters of K₂CO₃ and Li₂CO₃ with space group of *P6₃/mmc* are summarized in Table. R2. The significant difference between Li₂CO₃ and K₂CO₃ lies in the lattice parameters, M-O bond length, band gaps and sizes of interstitial sites.

Table. R2. Structure details of K₂CO₃, Li₂CO₃ and Li₂O₂.

P6₃/mmc	Lattice parameters			Formation energy (eV)	Density (g/cm ³)
	a	b	c		
K ₂ CO ₃	5.48	5.48	8.03	-2.12	2.21
Li ₂ CO ₃	4.65	4.65	5.38	-2.19	2.43
Li ₂ O ₂	3.24	3.24	8.29	-1.65	2.26

Different lattice parameters might result in the construction of carbonate/Li₂O₂ heterostructure with different well-matched lattice planes of carbonate and Li₂O₂. The Li-O bond length in Li₂CO₃ is similar with that in Li₂O₂, while the K-O bond length in K₂CO₃ is much longer than the Li-O in Li₂O₂ (Fig. R18). This difference might result in the construction of K₂CO₃/Li₂O₂ heterostructure with abundant interfacial defects. The smaller band gap of K₂CO₃ than Li₂CO₃ might be beneficial for the construction of heterostructure with better electronic conductivity. The presence of interstitial sites in K₂CO₃ with larger size is beneficial for the Li⁺ migration with enhanced solid-solution behavior.

Fig. R18. Crystal structure of a, Li_2CO_3 and b, K_2CO_3 with $P6_3/mmc$ space group. (Fig. R18 is added as Fig. S17 in **Supporting information**.)

4) There should be a band diagram, at least included in the SI, for the band levels for K_2CO_3 , Li_2O_2 , Li_2CO_3 based on the DFT calculations. Again, please incorporate U correction to ensure this is done correctly.

Response:

We highly appreciate the reviewer's constructive comments and professional suggestion. The band diagrams of K_2CO_3 , Li_2O_2 , Li_2CO_3 and $\text{K}_2\text{CO}_3/\text{Li}_2\text{O}_2$ heterostructure are calculated by density of states using GGA+ U functional with $U=6$ eV and added in the **Supporting Information**.

Fig. R16. Calculated densities of states (DOS), using GGA+U with $U=6$ eV, for a, Li_2O_2 ; b, $\text{K}_2\text{CO}_3/\text{Li}_2\text{O}_2$ heterostructure; c, Li_2CO_3 ; d, K_2CO_3 . (Fig. R16 is added as Fig. S16 in Supporting information.)

In summary, we highly appreciate the reviewer's professional suggestions and constructive comments. Following the detailed instruction, we carefully read through relevant literature and tried our best to understand the $\text{K}_2\text{CO}_3/\text{Li}_2\text{O}_2$ heterostructure.

REVIEWERS' COMMENTS:

Reviewer #1 (Remarks to the Author):

My comments have been well addressed. Now, it is suitable for publication.

Reviewer #2 (Remarks to the Author):

The authors have done an excellent job of addressing the reviewers concerns. They have taken the feedback holistically and improved the manuscript in many ways. The new version is suitable for publication.